# Yeast Display Reveals Plentiful Mutations That Improve Fusion Peptide Vaccine-Elicited Antibodies Beyond 59% HIV-1 Neutralization Breadth

**DOI:** 10.3390/vaccines13111098

**Published:** 2025-10-27

**Authors:** Camila T. França, Sergei Pletnev, Bharat Madan, Phinikoula S. Katsamba, Krisha McKee, Nicholas C. Morano, Baoshan Zhang, Fabiana Bahna, Tatsiana Bylund, Bob C. Lin, Mark K. Louder, Seetha Mannepalli, Rajani Nimrania, Sijy O’Dell, Nicole A. Doria-Rose, Peter D. Kwong, Lawrence Shapiro, Zizhang Sheng, Tongqing Zhou, Brandon J. DeKosky

**Affiliations:** 1The Ragon Institute of Massachusetts General Hospital, Massachusetts Institute of Technology, and Harvard University, Cambridge, MA 02139, USA; cfranca@mit.edu (C.T.F.); bharat.85.monu@gmail.com (B.M.); rajaninimrania10@gmail.com (R.N.); 2Department of Chemical Engineering, Massachusetts Institute of Technology, Cambridge, MA 02139, USA; 3Vaccine Research Center, National Institute of Allergy and Infectious Diseases, National Institutes of Health, Bethesda, MD 20892, USA; sergei.pletnev2@nih.gov (S.P.); mckeek@mail.nih.gov (K.M.); baoshan.zhang@nih.gov (B.Z.); tatsiana.bylund@nih.gov (T.B.); bob.lin@nih.gov (B.C.L.); mlouder@mail.nih.gov (M.K.L.); odells@mail.nih.gov (S.O.); nicole.doriarose@nih.gov (N.A.D.-R.); pdk3@cumc.columbia.edu (P.D.K.); tzhou@mail.nih.gov (T.Z.); 4Department of Pharmaceutical Chemistry, University of Kansas, Lawrence, KS 66047, USA; 5Department of Biochemistry and Molecular Biophysics, Columbia University, New York, NY 10032, USA; ps2396@cumc.columbia.edu (P.S.K.); nm3303@cumc.columbia.edu (N.C.M.); fb2019@cumc.columbia.edu (F.B.); sn2416@columbia.edu (S.M.); lss8@columbia.edu (L.S.); 6Zuckerman Mind Brain Behavior Institute, Columbia University, New York, NY 10027, USA; 7Aaron Diamond AIDS Research Center, Vagelos College of Physicians and Surgeons, Columbia University, New York, NY 10032, USA; zs2248@cumc.columbia.edu; 8Department of Chemical Engineering, University of Kansas, Lawrence, KS 66045, USA

**Keywords:** antibody improvement, broadly neutralizing antibody, epitope, fusion peptide, HIV-1 vaccine, paratope, yeast display

## Abstract

**Background/Objectives**: Vaccine elicitation of antibodies with high HIV-1 neutralization breadth is a long-standing goal. Recently, the induction of such antibodies has been achieved at the fusion peptide site of vulnerability. Questions remain, however, as to how much anti-fusion peptide antibodies can be improved and whether their neutralization breadth and potency are sufficient to prevent HIV-1 infection. **Methods**: Here, we use yeast display coupled with deep mutational screening and biochemical and structural analyses to study the improvement of the best fusion peptide-directed, vaccine-elicited antibody, DFPH_a.01, with an initial 59% breadth. **Results**: Yeast display identified both single and double mutations that improved recognition of HIV-1 envelope trimers. We characterized two paratope-distal light chain (LC) mutations, S10R and S59P, which together increased breadth to 63%. Biochemical analysis demonstrated DFPH-a.01_10R59P-LC, and its component mutations, to have increased affinity and stability. Cryo-EM structural analysis revealed elbow-angle influencing by S10R-LC and isosteric positioning by S59P-LC as explanations for enhanced breadth, affinity, and stability. **Conclusions**: These results, along with another antibody with enhanced performance (DFPH-a.01_1G10A56K-LC with 64% breadth), suggest that mutations improving DFPH_a.01 are plentiful, an important vaccine insight.

## 1. Introduction

Broadly neutralizing antibodies (bNAbs) against HIV-1 isolated from HIV-infected donors have been proposed as templates for vaccine development [1,2,3,4]. Immunization studies in macaques have also yielded potent bNAbs, such as LJF-0034, which recognizes the CD4-binding site and neutralizes approximately 70% of an 84-virus panel [5]. Other examples include the interface-directed antibody 1C2, with 87% neutralization breadth [6], and fusion peptide (FP)-specific antibodies that have been successfully elicited in multiple animal models, including mice, guinea pigs, and rhesus macaques [7,8]. The best of the FP-directed antibodies, DFPH-a.01, elicited in macaques, has 59% neutralization breadth on a cross-clade 208-strain panel [8]. Simian–human immunodeficiency virus (SHIV) boosting of FP-primed macaques increased serum neutralization breadth to 45–77% with a geometric mean potency of ~1:100 ID_50_ [9], levels that might prevent HIV-1 infection if achieved by vaccination alone [9]. Questions remain, however, about the breadth limit of FP-directed antibodies, as the template antibody for the FP site, naturally elicited antibody VRC34.01, has only ~50% breadth [10], which is lower than the breadths of template antibodies identified against other HIV-1 sites of vulnerability. For example, antibodies from natural infection show breadth that often exceeds 90% against the aforementioned CD4-binding site. Could vaccine-elicited FP-directed antibodies of higher breadth (>60%) be obtained? Or is there an intrinsic breadth limit? And how rare would such breadth-improving mutations be?

B cells sample mutations at varying rates during antibody development in vivo. Many of the possible mutations are rare, and only a small fraction of possible single mutations are effectively sampled in vaccine-animal models. In contrast, a yeast antibody library display paired with DNA site-saturation mutagenesis (SSM) and next-generation sequencing (NGS) can efficiently sample all single antibody mutations in large-scale affinity studies [11,12,13]. Yeast display thus provides a valuable platform to explore questions related to HIV antibody engineering and provides a mechanism to map potential mutational pathways based on affinity to diverse HIV-1 Env trimeric antigen probes. These library-scale affinity studies can evaluate rare mutations not sampled efficiently in vivo and can also explore combinatorial strategies that include both rare and commonly sampled mutations. This provides a system to address important questions about mechanisms of antibody improvement and to identify structural pathways that enhance the performance of vaccine-elicited HIV-1 broadly neutralizing antibodies.

In this study, we identified mutations that improved the breadth of the best FP-directed antibody, DFPH-a.01, which has 59% neutralization breadth with a geometric mean IC_50_ of 3.12 µg/mL [8]. In addition to being the best vaccine-elicited antibody, this antibody is also a member of a reproducible multi-donor antibody class [14,15,16], with antibody lineages with similar recognition and ontogeny being elicited in multiple macaques (including DFPH, 0PV, DJ85, GP6Z, and TRNM) [8,17]. Thus, a breadth or potency limit for DFPH-a.01 represents not only a limit for this particular antibody but also extends to a reproducible antibody class that makes up more than half of the broad FP-directed antibodies elicited in monkeys thus far. Here, we created single- and multi-mutation libraries and screened them for enhanced affinity against HIV Env trimers with diverse FPs. We identified antibodies with improved affinity, which we assessed for neutralization. With two improved multi-mutation variants, we analyzed biochemical stability and affinity to Env trimer. To provide atomic-level information, we also solved the structures of the two improved variants in complex with Env trimer. Overall, we found that we could enhance the neutralization breadth of DFPH-a.01 to 64%, that improving mutations were often rare or distal from the region of the antibody contacting the antigen, and that improving mutations were generally plentiful. These findings provide key insights to support the induction of broadly neutralizing immunity at the FP site of vulnerability.

## 2. Materials and Methods

### 2.1. Construction of DFPH-a.01 Variant Antibody Libraries

Site-saturation mutagenesis (SSM) libraries were constructed using primers encoding degenerate codons (NNK or MNN) to allow substitution by all 20 amino acids at each position of the variable heavy (VH) and light (VL) chains [11,12,13,18]. The resulting libraries were inserted into yeast surface display plasmids carrying FLAG and auxiliary expression cassettes, introduced into AWY101 cells, and expanded to over 2 × 10^6^ transformants to achieve full coverage [11,12,13,19].

To generate multi-mutation variants, enriched populations of single mutants with high affinity were sorted and further diversified using another round of SSM and DNA shuffling [11,12,13]. Four libraries of multi-mutants were generated. Library 1 contained randomized pairs of single-mutant VH and single-mutant VL chains (combinatorial, VH-SSM:VL-SSM). Library 2 was created by DNA shuffling of Library 1, in which template DNA was fragmented with DNAseI, reassembled, and reamplified (shuffled, VH shuffled/VL shuffled). Libraries 3 and 4 used an additional round of SSM on VH (VH-multi, VH-re-SSM:VL-SSM) or VL (VL-multi, VH-SSM:VL-re-SSM) (Appendix A). All plasmid DNA isolation and amplification from sorted yeast cells were performed as described [11,12,13].

### 2.2. Functional Library Screening

Yeast transformants were grown in induction medium containing 20 g/L galactose, 6.7 g/L yeast nitrogen base, 5 g/L casamino acids, 5.4 g/L Na_2_HPO_4_, and 8.6 g/L NaH_2_PO_4_·H_2_O (SGCAA, TEKnova, Hollister, CA, USA) supplemented with 2 g/L dextrose (SGDCAA). Cultures were incubated for 36 h at 20 °C with shaking at 225 rpm to promote antibody fragment (Fab) surface expression. Following induction, cells were washed and labeled with anti-FLAG–FITC antibody (clone M2, F4049; Sigma-Aldrich, Burlington, MA, USA) to assess Fab display levels.

Biotinylated HIV BG505 SOSIP trimers engineered to include one of three FP variants: AVGIGAVF (FP8v1, native BG505), AIGLGAMF (FP8v3), or AVGIGAMI (FP8-Thai) were used as antigenic probes and tested at 10 nM (FP8v1) or 50 nM (FP8v3, FP8-Thai). Yeast-displaying Fabs were stained with both anti-FLAG–FITC and antigen probes (streptavidin-PE label, 21388, Thermo Scientific, Waltham, MA, USA). Sorting was performed on a SONY MA900 cell sorter to separate Fab-positive cells into low-, mid-, and high-affinity populations, with four sequential enrichment rounds [11,12,13] (Appendix A). At least 3 × 10^7^ cells were analyzed in the initial round. A subset of Fab-positive yeast was retained prior to selection to monitor baseline variant distribution. After each sort, yeast was grown in SDCAA recovery medium (20 g/L dextrose, 6.7 g/L yeast nitrogen base, 5 g/L casamino acids, 10.4 g/L trisodium citrate, 7.4 g/L citric acid monohydrate, pH 4.5) for 24–48 h at 30 °C with shaking at 225 rpm.

### 2.3. Next-Generation Sequencing and Analysis

After each round of sorting, plasmid DNA was isolated from yeast, and VH and VL regions were amplified using barcoded primers [20,21,22]. Sequencing was carried out on an Illumina 2 × 300 MiSeq platform (Illumina Inc., San Diego, CA, USA) Raw FASTQ files were filtered with the FASTX Toolkit (v0.0.14, http://hannonlab.cshl.edu/fastx_toolkit/ accessed on 31 July 2025) to retain reads with quality scores ≥30 in at least 90% of bases. Filtered data were processed as previously described to recover in-frame antibody sequences [11,12,13].

Processed reads were aligned to the DFPH-a.01 reference sequence using USEARCH [23], and amino acid substitutions were annotated based on deviations from the template [11,12,13,24]. Mutation frequencies were used to calculate enrichment ratios (ERs) and compared across sequential selection rounds. Each sequence was assigned to high-, medium-, or low-affinity categories according to prevalence and ER trends across the dataset, as described previously [11,12,13,24].

### 2.4. Antibody Expression and Purification

Recombinant IgGs were produced as described [11,13]. Codon-optimized VH and VL genes were cloned into the VRC8400 vector and co-transfected into Expi293F cells (2.5 × 10^6^/mL; Thermo Fisher Scientific, Waltham, MA, USA) using Turbo293 reagent (SPEED BioSystems, Gaithersburg, MD, USA). Cultures were incubated at 37 °C with 9% CO_2_ and agitation at 120 rpm. After five days, supernatants were harvested, and antibodies were purified by Protein A chromatography (GE Healthcare, Chicago, IL, USA), neutralized with 1 M Tris-HCl (pH 8.0), and analyzed by SDS-PAGE for purity.

### 2.5. Virus Neutralization

Neutralization activity of recombinantly expressed monoclonal antibodies was evaluated against a panel of 20 viral strains using luciferase-based entry assays. Antibodies were serially diluted fivefold from 500 µg/mL and mixed (50 µL) with reporter virus stocks. After 1 h incubation at 37 °C, 20 µL of TZM-bl cells (0.5 × 10^6^ cells/mL; NIH AIDS Reagent Program, Bethesda, MD, USA) were added and cultured overnight. The following day, 130 µL of complete DMEM was added, and cells were incubated for another 24 h. On day 3, luciferase activity was measured as relative light units to quantify infection. IC_50_ and IC_80_ values were calculated by nonlinear regression (Hill-slope model) as previously described [11,13]. To assess antibody activity across global FP diversity, neutralization was measured against a 208-virus HIV-1 Env pseudotype panel using automated 384-well microneutralization assays as previously described [25].

### 2.6. Affinity Measurements by Surface Plasmon Resonance

Binding interactions were analyzed by surface plasmon resonance (SPR) on a Biacore T200 instrument (Cytiva, Marlborough, MA, USA) using a Series S CM5 sensor chip (Cytiva, Marlborough, MA, USA). Experiments were carried out at 25 °C in HBS-P buffer (10 mM HEPES, pH 7.4, 150 mM NaCl, and 0.05% *v*/*v* Tween-20).

Antibody 2G12 (IgG) was immobilized on all surfaces using amine-coupling chemistry to ~10,000 response units (RUs); this antibody was used to tether BG505 DS-SOSIP to the chip surface at approximately 400 RU. Binding of the wild type DFPH-a.01 and its respective mutants was tested at five concentrations ranging from 1.11 to 90 nM, which were prepared in a running buffer using a three-fold dilution series. Binding cycles consisted of a 120 s association phase and 900 s dissociation phase at a flow rate of 50 µL/min, followed by a 60 s regeneration step using 3 M MgCl_2_ at 30 µL/min. A surface immobilized with 2G12 IgG was used as a reference surface to subtract bulk refractive index changes, and buffer blanks, in which a running buffer without Fab was flowed over the BG505 DS-SOSIP surfaces, were used to subtract systematic noise drift. Binding responses were globally fit to a 1:1 interaction model using Scrubber 2.0 to determine the kinetic parameters k_a_ and k_d_ as well as the equilibrium binding constant K_D_ for each interaction.

### 2.7. Stability Measurements

The Fab of wild type DFPH-a.01 and select mutants were assessed for stability using a TychoTM NT.6 to measure the ratio of intrinsic tryptophan fluorescence at 350 and 330 nm. All Fabs were diluted to 1 mg/mL, loaded into capillaries, and melted over a temperature range of 30–95 °C. Each Fab was assessed in experimental triplicate, and the data was normalized to the starting 350/330 ratio.

### 2.8. Cryo-EM Structure Determination

BG505 DS-SOSIP HIV-1 envelope trimer and DFPH-a.01_10R59P-LC Fab were concentrated to 6.5 mg/mL and 8.0 mg/mL, respectively, and mixed at a molar ratio of 1:1.3. To prevent preferred orientation, the complex was supplemented with 0.1 mM of n-Dodecyl β-D-maltoside (DDM; final concentration).

Quantifoil R 2/2 gold grids were treated by glow discharge using a PELCO easiGlow (Ted Pella, Inc., Redding, CA, USA) unit (0.39 mbar, 20 mA, 30 s). A 2.7 µL aliquot of sample was applied to each grid and vitrified in liquid ethane with a Vitrobot Mark IV (ThermoFisher Scientific, Waltham, MA, USA) under controlled conditions (4 °C, 95% humidity, blot force −5, blot time 1.5–3 s). Cryo-EM data were acquired at the National Cryo-Electron Microscopy Facility (NICE, Frederick, MD, USA) using an FEI Titan Krios microscope (ThermoFisher Scientific, Waltham, MA, USA) equipped with a Gatan K2 Summit direct electron detector (Gatan Inc., Pleasanton, CA, USA), operated in super-resolution mode (pixel size 0.415 Å before binning). Data collection was performed with SerialEM v4.1 [26] (Appendix A). Cryo-EM reconstruction has been performed with CryoSPARC v3.3 [27]. Movies were aligned, dose-weighted, and binned to 0.83 Å using patch motion correction, and the micrograph contrast transfer function (CTF) parameters were estimated using patch CTF estimation. Particles were picked using the blob picker, extracted from the micrographs, and subjected to 2D classification followed by the selection of the best classes. Ab initio reconstruction and heterogeneous, homogeneous, and non-uniform refinement jobs were run in C1. Local resolution was determined with the local resolution module in CryoSPARC.

To obtain initial atomic models, the complex with coordinates of the BG505 DS-SOSIP trimer (PDB 8EUV) [11] and the AlphaFold2 [28,29] model of the DFPH-a.01_10R59P-LC Fab was docked into corresponding parts of the cryo-EM map in UCSF Chimera v1.19 [30]. Atomic models were refined by alternating rounds of model building in Coot v0.9.7 [31,32] and real-space refinement in Phenix v1.21 [33]. Structure validation was performed with Molprobity [34,35] and the PDB validation server. The analysis of HIV-Fab interfaces was performed with the EMBL PISA server [36]. Summaries of model refinement statistics and quality assessment for cryo-EM reconstructions are given in Appendix A. Structure figures were generated with UCSF Chimera v1.19 [30], ChimeraX v1.1 [37], and Pymol v1.8.6.2 (Schrodinger, Inc., New York, NY, USA, https://pymol.org/).

### 2.9. Quantification and Statistical Analysis

IC_50_ values were determined using GraphPad Prism v10.5 (Siemens, Washington, DC, USA) by fitting experimental data to a four-parameter logistic regression model with the lower bound fixed at zero. Flow cytometry data processing and figure generation were performed with FlowJo software (v10.8) [38].

## 3. Results

### 3.1. Precision Yeast Display Antibody Engineering of DFPH-a.01

We selected the vaccine-elicited FP-directed antibody with the highest neutralization breadth, antibody DFPH-a.01, which has an IC_50_ neutralization breadth of 59% on a 208-strain panel [8], to investigate features of potential mutations that could further improve its efficacy (Figure 1A). We performed deep mutational scanning to evaluate the functional impact of every single amino acid (aa) mutation on the heavy (VH) and light (VL) chains of DFPH-a.01 [11,12,13,18]. SSM variant libraries were cloned into yeast surface display and stained with BG505 SOSIP trimer antigens for FACS. A >250-fold theoretical library coverage was maintained throughout all cloning steps to ensure robust data collection. SSM libraries were screened for their affinity to three BG505 DS-SOSIP.664 HIV-1 Env trimers with a distinct globally circulating FP variant: BG505_FP8-v1, BG505_FP8-v3, and BG505_FP8-Thai (Figure 1A) [11]. Single-mutant libraries for heavy and light chains (VH-SSM and VL-SSM, respectively) were analyzed by flow cytometry and gated based on the ratio of surface Fab expression to HIV-1 antigen binding. The established sorting strategy fractionated the antibody library into three different populations based on binding affinity (Figure 1B). Yeast-expressing mutations detrimental to Env binding were enriched in the low-affinity gate, whereas mutations with no significant impact or with binding comparable to the template DFPH-a.01 were sorted into the medium-affinity gate. Any mutations with enhanced affinity were enriched in the high-affinity gates. After four rounds of affinity-based sorting, high-affinity populations showed clear phenotypic binding improvements in both the VH-SSM and VL-SSM libraries, for all three HIV-1 antigens used in the study (Figure 1B and Appendix A).

Sorted libraries were analyzed by NGS to bioinformatically track enriched mutant sequences based on the composition of sorted gates. We evaluated the impact of each mutation based on enrichment ratios across screening rounds. Among single amino acid mutations in the heavy chain, 45% showed a detrimental effect on HIV-BG505_FP8-v1 binding affinity. Another 38% led to an affinity similar to the template antibody. The remaining 0.6% of single mutations had a beneficial effect (Figure 2A and Appendix A). Similarly, 42%, 47%, and 0.8% of the light chain mutations showed deleterious, neutral, or enhancing effects on binding affinity, respectively (Figure 2A and Appendix A). The top 10 single mutants (based on enrichment ratios) were selected for further analysis.

### 3.2. Multi-Mutation Library Generation and Screening

To identify potentially synergistic multi-mutation combinations, we performed another round of SSM paired with DNA shuffling of enriched single mutants (Figure 1A). We expected that multi-mutation combinations would outperform single mutations in yeast display [11,12,13]. Four multi-mutation libraries were generated by pooling the high-affinity single-mutant libraries isolated after three rounds of enrichment. Libraries were enriched for high-affinity binders, and after three rounds of sorting, the libraries showed enhanced trimer recognition compared to the template antibody against all three antigens (Figure 2B). NGS data were mined for potentially beneficial combinations of mutations, and the top six light chain multi-mutants were selected for expression and characterization based on persistence in multi-mutation rounds 2 and 3 against any of the three antigens and highest enrichment ratios in round 2 or round 3, in some cases containing at least one light chain mutation known to enhance neutralization from single mutation tests (Appendix A).

### 3.3. HIV-1 Neutralization Analysis of Mutational Variants Identified in Yeast Display

Variants with affinity-improving mutations identified by NGS were expressed as IgG1 to evaluate the potential effects on neutralization potency and breadth. We first assessed single-mutation DFPH-a.01 variants for HIV-1 against a panel of 20 viral isolates that were selected based on predictive capacity for broad FP-specific neutralization. The panel included five isolates that encoded FP8-v1 (AVGIGAVF), two with FP8-v2 (AVGLGAVF), two with FP8-v3 (AIGLGAMF), two with FP8-v4 (AVGTIGAMF), four with FP8-Thai (AVGIGAMI), one with FP8-v6 (AVGIGAMF), and four with other FP sequences (Appendix A).

Ten DFPH-a.01 single-mutation variants were selected based on high-affinity enrichment against at least one FP8 antigen after round 2 and/or 3 (five mutations in the heavy chain, five mutations in the light chain) and were assayed for HIV-1 neutralization in the 20-virus panel. Of these, several showed some enhanced breadth in the 20-virus panel (Figure 2C and Appendix A). The most potently improved single amino acid mutant, S10R-LC, showed 1.7-fold better potency against viruses carrying the FP variant FP8-v1 (geomean IC_50_ = 2.98 µg/mL compared to DFPH-a.01 geomean IC_50_ = 5.10 µg/mL), 1.4-fold better potency against FP8v2 (geomean IC_50_ = 6.29 µg/mL versus 8.91 µg/mL), and 1.3-fold better potency against FP8-Thai variants (geomean IC_50_ = 7.96 µg/mL versus 10.5 µg/mL). These improvements corresponded to an increase in 20-virus panel IC_50_ breadth from 75% to 85% (Figure 2C and Appendix A).

Next, we analyzed selected variants from the multi-mutation screens to identify synergistic effects (Appendix A). On the 20-strain panel, the top-performing variant DFPH-a.01_10R59P-LC (with S10R-LC and S59P-LC) showed a 2.7-fold improved neutralization potency (geomean IC_50_ = 2.15 µg/mL compared with DFPH-a.01 geomean IC_50_ = 5.87 µg/mL) (Figure 2D and Appendix A). Two other variants (D1G_S10A-LC and D1G_S10A_T56K-LC) also showed improvements in IC_50_ and IC_80_ neutralization data. Based on these results, we sought to better characterize DFPH-a.01_10R59P-LC on a larger panel and to learn more about the mechanisms of its improvement.

### 3.4. Breadth and Potency Analysis of DFPH-a.01_10R59P-LC

Based on the 20-virus panel data, DFPH-a.01_10R59P-LC was selected for evaluation against a broader 208-strain pseudovirus panel that is globally representative of HIV-1 strain diversity [7,11,13] (Appendix A). We found that in addition to incremental potency improvements (Figure 3A,B, Appendix A), these mutations improved the IC_50_ neutralization breadth from 59% to 63%. This level of neutralization was comparable to the highest breadth identified to date among non-engineered monoclonal antibodies (PGT151) [7,8,10,11,13,17,39,40,41] (Figure 3C,D and Appendix A). DFPH-a.01_10R59P-LC also showed gain-of-function neutralization with the FP8-v3 sequence and against two strains encoding the FP-Thai sequence (50 µg/mL cutoff) (Figure 3A–C and Appendix A).

### 3.5. Surface Plasmon Resonance and Thermal Stability Analyses

To evaluate how the introduced mutations affected binding affinity, we performed surface plasmon resonance analyses to compare the affinities of DFPH-a.01 wild type and variant antibodies for the soluble, prefusion-stabilized Env trimers (BG505 DS-SOSIP). The double mutant DFPH-a.01_10R59P-LC had a K_D_ of 113 pM, roughly twice the binding affinity of wild type, which had a K_D_ of 209 pM (Figure 4A), suggesting higher affinity as the basis for its increased neutralization breadth. Interestingly, the constituent individual mutations actually had higher affinities than the multi-mutation variant; DFPH-a.01_10R-LC had a K_D_ of 97 pM and DFPH-a.01_59P-LC had a K_D_ of 113 pM, indicating that the two underlying mutations were not additive for affinity.

To characterize stability, we monitored intrinsic fluorescence of tryptophan by assessing the ratio of fluorescence at 350 nm versus 330 nm over a temperature range of 30–95 °C. We observed both double and constituent single mutations to have increased melting temperatures. Interestingly, single mutations enhanced stability only marginally (0.3 °C and 1.0 °C), whereas the double mutant had a melting temperature that was 2.5 °C higher (Figure 4B). Thus, in contrast to affinity, the two single mutations appeared to synergize with respect to increased stability.

### 3.6. Cryo-EM Structure of Fab DFPH-a.01_10R59P-LC with BG505 DS-SOSIP Env Trimers

To investigate the structural basis for the observed improvements in affinity and stability of DFPH-a.01_10R59P-LC, we purified its antigen-binding fragment, generated complexes with BG505 DS-SOSIP Env trimers, and collected single particle cryo-EM data on a Titian Krios microscope (ThermoFisher Scientific, Waltham, MA, USA). From 347,691 particles, we obtained a 3.0 Å resolution reconstruction, for which we built an atomic-level model comprising the component Fab and Env trimers (Figure 5A and Appendix A). We had previously determined the structure of another lineage member (DFPH-a.15, which had 46 amino acid mutations versus DFPH-a.01) and observed highly similar recognition (Appendix A). In terms of the two mutations, both were distal from the paratope, with 10R-LC being 23.4 Å distal and 59P-LC being 9.1 Å distal (Figure 5B).

The S10R-LC mutation was located at the interface between the variable domains and the constant domains, at the elbow of the Fab (Figure 6A). While the constant regions of most other members of the class have not been defined, the structure of the constant regions has been defined for the vaccine-elicited, SHIV-infection boosted TRNM-b.01 antibody, which has a serine at VL position 10. Comparison with TRNM-b.01 revealed residue 10 to influence the elbow angle, with a 15-degree difference between DFPH-a.01_10R59P-LC and TRNM-b.01 (Figure 6B). Analysis of the structural context for the S59P mutation revealed its ability to provide isosteric positioning of T56 CDR-L2 contact with Env trimers (Figure 6C).

### 3.7. A Second Yeast Display-Identified Antibody, DFPH-a.01_1G10A56K-LC, with 64% Neutralization Breadth

To gain insight into the prevalence and the increase in neutralization by other DFPH-a.01 variants identified by yeast display, we characterized a second multi-mutation variant, DFPH-a.01_1G10A56K-LC (Appendix A). This antibody showed slightly higher breadth at 64% than DFPH-a.01_10R59P-LC (Figure 7A). This second antibody showed a similar FP neutralization profile, and it was able to neutralize v1 and v2 variants of FP8 (Figure 7B). SPR analysis revealed a 3-fold increase in K_D_ (to BG505 DS-SOSIP), related mostly to the increased association rate (Figure 7C). A slight increase in stability was also observed (Figure 7D). Overall, several single-, double-, and multi-mutation variants were found to have increased neutralization breadth on the 20-isolate panel, and several of these showed increased potency (Figure 7E).

### 3.8. Comparative Analysis of Improving FP Broadly Neutralizing Antibody Mutations Across Macaque, Mouse, and Human Studies

To understand generalizable features for antibody improvement against the FP epitope, we compared the data collected here for DFPH-a.01 (a macaque vaccine-elicited antibody) with prior data from yeast display campaigns with VRC34.01 [11] (a human, naturally elicited antibody) and with vFP16.02 [13] (a mouse, vaccine-elicited antibody) (Figure 8A). Each of these three antibodies is among the very best anti-FP broadly neutralizing antibodies that have been isolated in their respective species. Across all three species, we found that site-saturation mutagenesis and yeast display identified few highly beneficial mutations in the CDR3s. The most beneficial mutations identified were commonly outside of CDRs (5/6 red mutations, Figure 8B) and sometimes occurred in paratope-distal locations, especially for vaccine-elicited antibodies. These paratope-distal substitutions improved DFPH-a.01 by allosterically rigidifying the Fab elbow (DFPH-a.01_S10R-LC) and increasing CDR loop flexibility (DFPH-a.01_S59P-LC). Similarly, deep mutational scanning of the murine broadly neutralizing antibody vFP16.02 showed that many beneficial mutations were framework changes distant from the paratope [13] (Figure 8B). In particular, the vFP16.02 light-chain S48K (S43K in Kabat numbering) mutation enhanced neutralization breadth (from ~28% to 37%) by altering the VH–VL orientation without directly contacting the FP [13]. In contrast, several mutations that improved the naturally elicited human antibody VRC34.01 made direct contact with the antigen, including VRC34.01_E2K_VH, which generated a new contact site [11].

We also sought to understand the prevalence of identified mutations in natural immune repertoires. For DFPH-a.01, which was isolated from an immunized macaque, we identified both frequent and rare improving mutations in yeast display: S59P-LC was common (13% of natural sequences), whereas S10R_LC_ was very rare (<0.1% of sequences). However, the three most important heavy-chain substitutions for the naturally elicited human broadly neutralizing antibody VRC34.01_mm28 (VH_E2K, VH_A33P, and VH_T58F) had either rare or moderate prevalence, with frequencies in humans of 0.01%, 0.1%, and 0.72%, respectively (Figure 8B).

While improving mutations that made direct contact with the antigen were sometimes observed in vaccine-elicited antibodies, we note that many promising single mutations did not make direct contact with the antigen; thus, of the 16 promising single mutants identified by yeast display, 14 of these (the majority) were not part of the paratope (Figure 8C). One of these, DFPH-a.01_S59P-LC, allowed for the isosteric positioning of nearby residues that did indeed contact the antigen. We note that for VRC34.01, the ratio of single mutations that were or were not part of the paratope is more balanced, with half the identified improving mutations contacting the epitope and half not (Figure 8C), suggesting that the ratio of contact versus non-contact improvements can depend on particulars of antibody and recognition. Overall, these data suggest diverse mechanisms by which contact and non-contact residues can contribute to improved FP-directed neutralization breadth and potency across species.

**Figure 8 vaccines-13-01098-f008:**
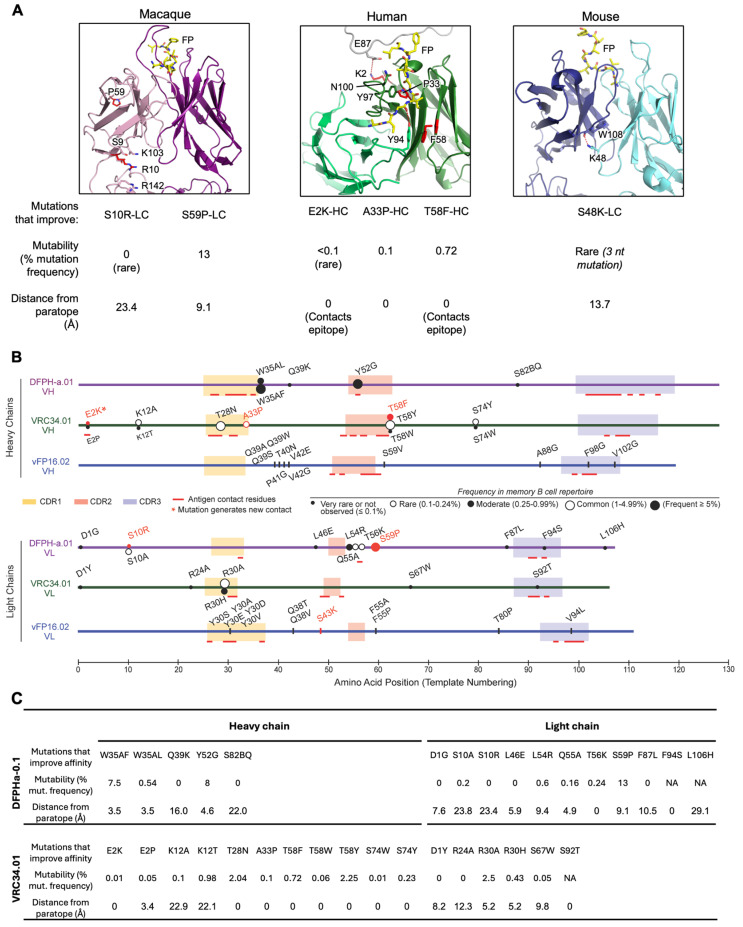
Paratope distance and mutation frequencies for yeast display-identified mutations that improve broadly neutralizing antibodies for macaque, mouse, and human. (**A**) Paratope-distal mutations that enhance vaccine-elicited antibodies isolated from macaque, human, and mouse. (**B**) Overview of mutations observed in anti-FP antibodies. Top (most protective) mutations are shown in red. Tick marks were used for vFP16.02 because mutation prevalence data is not available for mouse germline genes. * Indicates a new contact region established by the heavy chain mutation E2K. Only contact residues with BSA (Buried Surface Area) equal to/higher than 15 Å^2^ are indicated. Mutations indicated in Kabat numbering. (**C**) Mutation frequency and distance from the paratope for identified mutations. See also Appendix A.

## 4. Discussion

In this study, we used yeast display-affinity engineering to enhance the neutralization breadth and potency of the top macaque vaccine-elicited antibody against HIV-1 FP, DFPH-a.01. By further improving DFPH-a.01, we demonstrated here that vaccine-elicited antibodies against HIV-1 FP in macaques are not limited to a neutralization breadth of <60%. We identified multiple mutations that enhanced the sorting affinity to the Env trimer with diverse FPs. We characterized several of these mutations in depth, including two multi-mutant variants, DFPH-a.01_10R59P-LC and DFPH-a.01_1G10A56K, with breadths of 63% and 64% on a 208-strain panel, respectively. Overall, many individual mutations appeared to improve DFPH-a.01, suggesting that improvement mutations are plentiful for anti-FP antibodies of this reproducible antibody class.

Beyond breadth, our data also demonstrate meaningful improvements in potency. The DFPH-a.01_10R59P-LC variant showed ~2.7-fold enhanced IC_50_ potency in the 20-virus panel, and more modest but consistent potency gains across the 208-virus panel. These improvements are biologically meaningful, as macaque passive transfer studies indicate serum ID_50_ (inhibitory dilution required to inhibit 50% of infection) titers of ≥1:100 to be sufficient to prevent SHIV acquisition [17]. Even modest increases in antibody potency can lower the threshold concentration required for protection, enhancing the likelihood that vaccine-elicited antibodies achieve protective titers in vivo. Together, the combined improvements in breadth and potency highlight the translational impact of engineering FP-directed antibodies for vaccine development.

The high prevalence of mutations to improve DFPH-a.01 suggests the breadth/potency limit for DFPH-a class antibodies to be substantially higher than the 59% breadth of the current best vaccine-elicited antibody (DFPH-a.01). This is an important finding, as the breadth of the SHIV infection-boosted antibodies was highest at only 56% (for DJ85-a.01). Despite substantial in vivo affinity maturation (somatic hyper mutations-SHM-rates of 15–20%), the SHIV infection-boosted antibodies still did not appear to achieve that same level of maximal breadth. Likely, this reflects the manner by which these antibodies were selected and matured and suggests that the immunization process can be further optimized to maximize FP-directed neutralization breadth. Even modest increases in breadth may have meaningful biological implications when accompanied by corresponding potency gains. These findings suggest that raising the breadth of DFPH-a-class antibodies, together with the potency improvements observed here, can combine to enhance protective efficacy.

Overall, yeast display revealed that many mutations can improve the neutralization breadth of DFPH-a.01, the best of the DFPH-a class of ‘reproducible’ or ‘multi-donor’ antibodies. This suggests that the breadth limit for this class of FP-directed antibodies is considerably higher than 60%, and the 63–64% breadth achieved here is unlikely to represent a ceiling for FP-directed antibodies. These yeast display efforts revealed panels of additional beneficial mutations, many of which were not fully evaluated in this study, suggesting that additional engineering would continue to enhance both potency and breadth. Given the high conservation of the FP across HIV-1 isolates and the repeated elicitation of FP-directed lineages in multiple animals, there appears to be substantial evolutionary space for additional improvements. Nevertheless, whatever the breadth limit, the breadth of FP-directed antibodies engineered here already exceeds the target 50% level, which has been suggested as a minimal breadth level that would lead to an impactful vaccine.

Our mutational analyses also offer insights into rational immunogen design and germline-targeting strategies. Distal framework mutations, such as S10R-LC, which alters Fab elbow angle, and S59P-LC, which increases CDR loop flexibility, provide further evidence that non-paratope residues can strongly influence FP antigen recognition. These structural pathways suggest that immunogens could be designed to favor B cell lineages with the capacity to access such mutations, thereby shaping antibody maturation toward mutationally favorable frameworks. Indeed, rational immunogen design has demonstrated proof of principle for guiding antibody evolution through tailored antigen engineering [42,43,44,45]. Applying these principles to germline-targeting strategies may facilitate the elicitation of FP-directed antibodies with improved breadth and potency and would synergize well with recent structure-based FP immunogen designs. Interestingly, the most improving mutations for the macaque FP-directed antibody DFPH-a.01 emerged at paratope-distal framework positions, whereas beneficial mutations in human FP-directed antibodies (e.g., VRC34.01) were often concentrated at or near the paratope. These differences may reflect distinct starting points for affinity maturation and/or differences in the total duration of antigen exposure. In the macaque lineage, DFPH-a.01 began with relatively low baseline affinity and breadth, such that distal framework changes altering Fab elbow angle (S10R-LC) and CDR loop flexibility (S59P-LC) created structural preconditions that expanded recognition of diverse FP variants. In contrast, human FP bNAbs may have already achieved high degrees of paratope complementarity, so further improvements were driven by fine-tuning residues proximal to the antigen contact surface. These differences highlight that multiple mutational trajectories, either distal framework remodeling or direct paratope optimization, can support the evolution of increased breadth and potency in FP-directed bNAbs.

We note that yeast display is a system designed to explore protein–protein interaction affinities, and it does not fully recapitulate in vivo B cell evolution. SHM and selection occur inside germinal centers under complex intraclonal interaction dynamics based on factors such as antigen availability, competition among B cell clones, and the availability of T cell help. As a result, some mutations identified here may not be readily accessible to natural repertoires. Nevertheless, yeast display remains a powerful tool for uncovering structural mechanisms and mutational pathways to inform the next steps in rational vaccine design.

## Figures and Tables

**Figure 1 vaccines-13-01098-f001:**
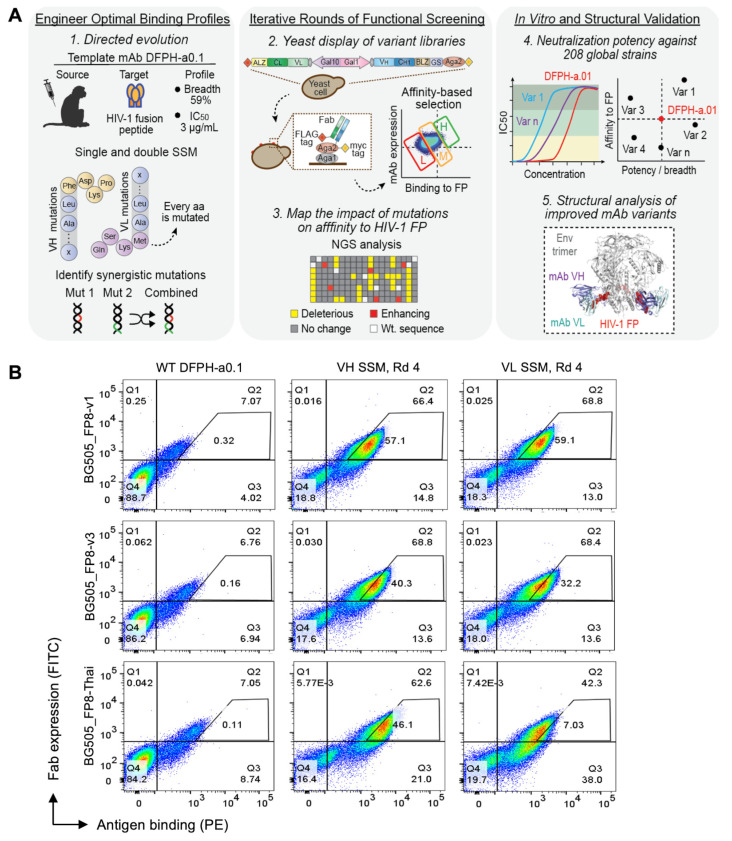
Precision antibody yeast display to evaluate mutational landscapes for the HIV-1 broadly neutralizing antibody, DFPH-a.01. (**A**) Site-saturation mutagenesis (SSM) introduced diversity into the DFPH-a.01 variable region genes, which were cloned and displayed as Fab libraries on yeast. Flow cytometry screened variants for binding to BG505 SOSIP Env trimers bearing distinct HIV-1 fusion peptide (FP8) sequences (FP8-v1, FP8-v3, and FP8-Thai). Sorted libraries were sequenced to map each mutation’s impact on affinity. High-affinity variants were expressed and evaluated in neutralization assays. Structural analysis revealed mutational mechanisms underlying improved breadth and potency. (**B**) Binding for the wild type (WT) antibody DFPH-a.01 compared to the enriched heavy (VH) and light (VL) chain libraries after four rounds of selection against three Env trimers. See also Appendix A.

**Figure 2 vaccines-13-01098-f002:**
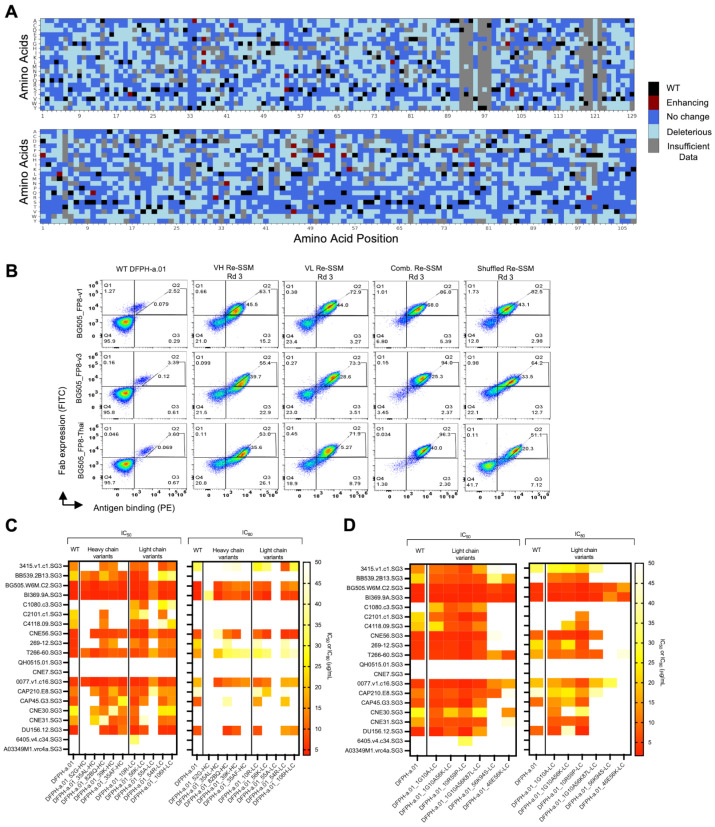
Bioinformatic analysis of library screening data reveals multiple single mutations that enhance HIV-1 neutralization potency and breadth, with multi-mutation screening revealing efficacious combinations. (**A**) Heat maps display the functional impact of single mutations on binding to HIV-1 BG505-FP8v1 Env. (**B**) Flow cytometry profiles show binding of wild type (WT) DFPH-a.01 compared to the enriched heavy (VH) and light (VL) chain libraries after three rounds of selection against three Env trimers. (**C**) Selected single mutants were expressed as IgG1 and evaluated in pseudovirus IC_50_ neutralization assays against a 20-virus panel; IC_50_ ≤ 50 µg/mL was used as the threshold for neutralization sensitivity. (**D**) Multi-mutation screening identified synergistic combinations that improved neutralization potency and breadth against diverse HIV-1 variants using the same IC_50_ threshold. See also Appendix A.

**Figure 3 vaccines-13-01098-f003:**
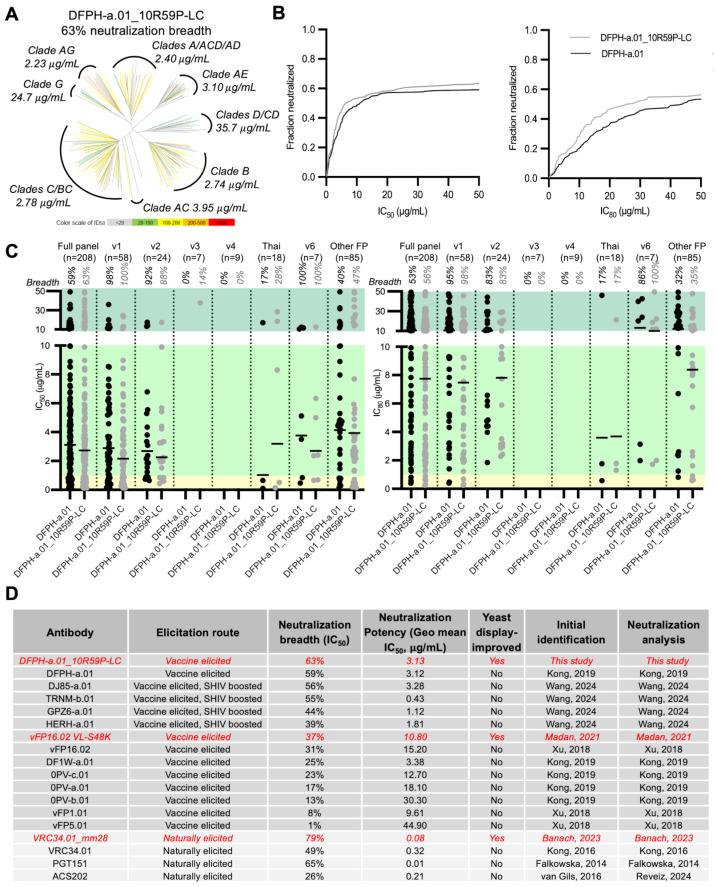
Assessment of DFPH-a.01_10R59P-LC on a 208-strain panel reveals enhanced neutralization potency and breadth. (**A**) Dendrogram analysis illustrates that the engineered antibody DFPH-a.01_10R59P-LC exhibits broader cross-clade recognition and improved neutralization against both DFPH-a.01-sensitive and -resistant pseudoviruses. Geometric mean potency is indicated. (**B**) Neutralization curves demonstrate progressive gains in potency and breadth relative to the parental antibody across the virus panel. (**C**) Results from a 208-strain panel confirmed expanded breadth of the engineered variant compared with the template, with viral strains organized by fusion peptide sequence. Neutralization sensitivity was defined by IC_50_ (**left**) or IC_80_ (**right**) ≤ 50 µg/mL. (**D**) Comparison of known anti-FP neutralizing antibodies. See also Appendix A.

**Figure 4 vaccines-13-01098-f004:**
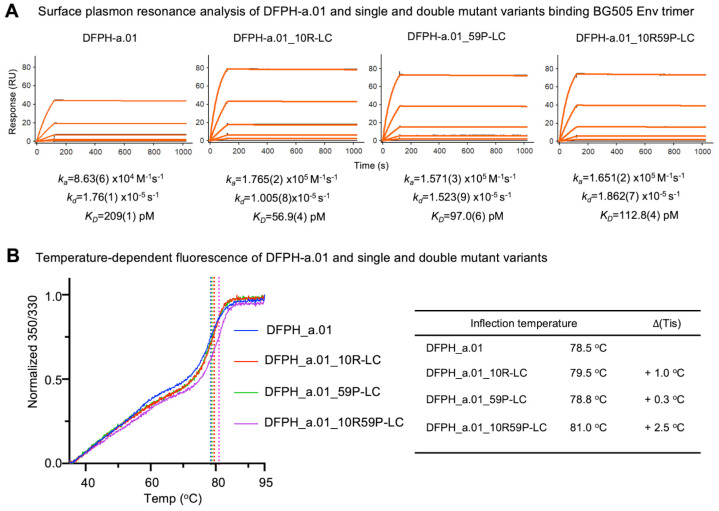
Surface plasmon resonance and temperature-dependent fluorescent analyses reveal DFPH-a.01_10R-LC to be 4-fold improved in affinity and DFPH-a.01_10R59P-LC to be improved by 2.5 degrees in stability. (**A**) Surface plasmon resonance analysis of DFPH-a.01 and the single and double mutants for binding to BG505 Env trimers. Binding analysis was performed using BG505 DS-SOSIP tethered to the sensor chip surface via 2G12, with DFPH-a.01 and its respective mutants tested at five concentrations ranging from 1.11 to 90.0 nM. Black traces represent experimental data, and red lines represent the fit to the 1:1 interaction model. Numbers in brackets show the error of the fit to the last significant digit. (**B**) DFPH_a.01, DFPH_a.01_10R-LC, DFPH_a.01_59P-LC, and DFPH_a.01_10R59P-LC were assessed for stability by monitoring the change to the 350/330 ratio from intrinsic fluorescence of tryptophan during melting. All three variants showed improved stability as compared to wild type DFPH_a.01. DFPH_a.01_10R-LC increased stability by 0.3 °C, while DFPH_a.01_59P-LC increased stability by 1.0 °C. The double mutant displayed increased stability by 2.5 °C, indicating possible structural synergy between the two mutations, as combined they enhance stability greater than either individual mutant. The results shown are averaged from three experimental replicates and normalized with the starting 350/330 ratio set to 0. See also Figure 5, Figure 6, Figure 8, and Appendix A.

**Figure 5 vaccines-13-01098-f005:**
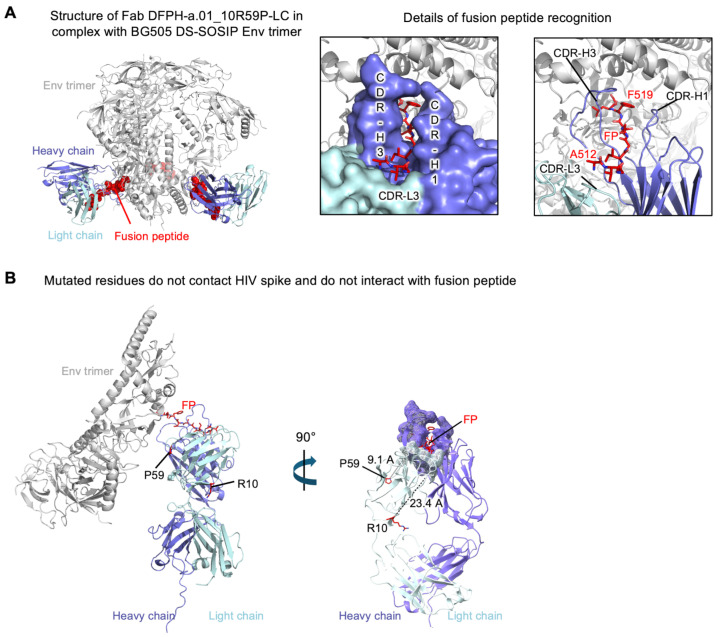
The cryo-EM structure of Fab DFPH-a.01_10R59P-LC with BG505 DS-SOSIP Env trimers reveals mutations that improve affinity and stability to be located distal from the fusion peptide-epitope. (**A**) Details of DFPH-a.01_10R59P-LC recognition. BG505 DS-SOSIP trimer is shown in gray, the fusion peptide is shown in red, and heavy and light chains of DFPH-a.01_10R59P-LC Fab are shown in purple and light cyan, respectively. Fusion peptide is threaded between the complementarity-determining region (CDR)-H1 and CDR-H3 with its N-terminus fixed by CDR-L3. (**B**) Location of mutations 10R and 59P in DFPH-a.01_10R59P-LC Fab. Neither R10 nor P59 is in contact with the antigen. The shortest distances to the paratope for R10 and P59 are 23.4 A and 9.1 A, respectively. See also Appendix A.

**Figure 6 vaccines-13-01098-f006:**
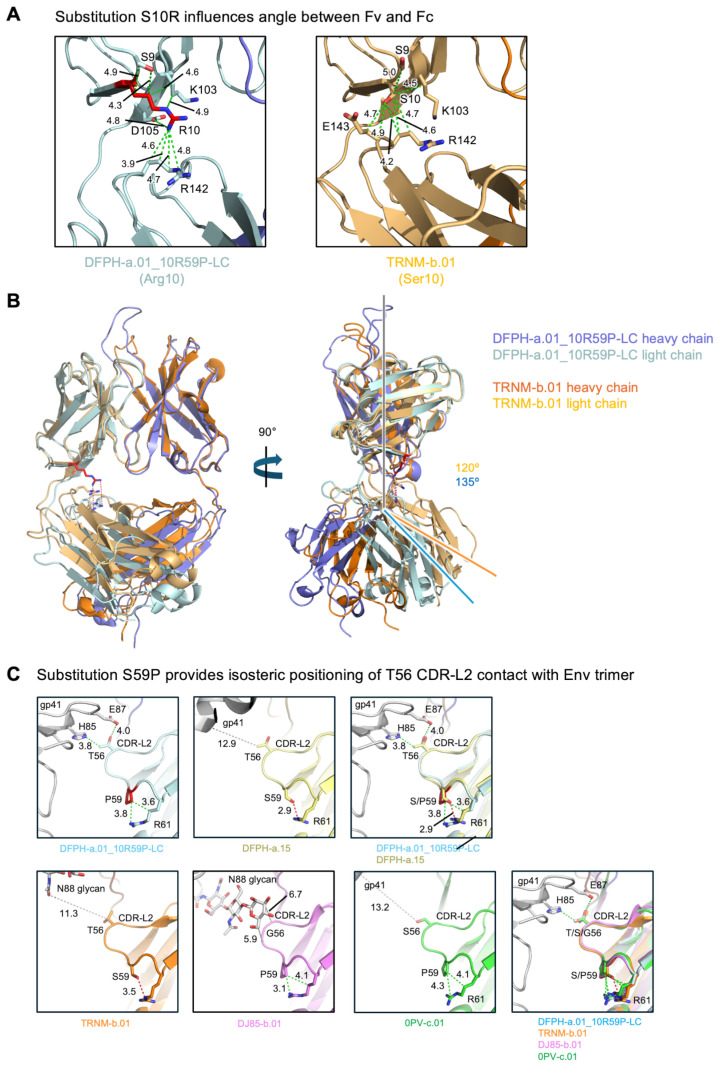
Cryo-EM structural explanations: S10R influences elbow angle and S59P enables isosteric positioning. (**A**) Details of residue 10 interactions in DFPH-a.01_10R59P-LC and in TRNM-b.01 highlighting the elbow angle difference of 135° versus 120°, respectively. The distances between 10R and other residues that are shorter than 5 Å are shown in green. (**B**) Overall view of Fabs, after variable domain superposition, highlighting different elbow angles for DFPH-a.01_10R59P-LC (which has 10R) and TRNM-1.b01 (which has 10S). (**C**) The conformation of CDR-L2 in DFPH-a.01_10R59P-LC, TRNM-b.01, DJ85-b.01, 0PV-c.01, and DFPH-a.15. The shortest antibody–antigen distances of less than and more than 5 Å are shown in green and gray, respectively. The CDR-L2 of DFPH-a.01_10R59P-LC forms a weak contact with H85 and E87 of the HIV trimer. The CDR-L2 of TRNM-b.01, DJ85-b.01, 0PV-c.01, and DFPH-a.15 do not interact with the HIV trimer. See also Appendix A.

**Figure 7 vaccines-13-01098-f007:**
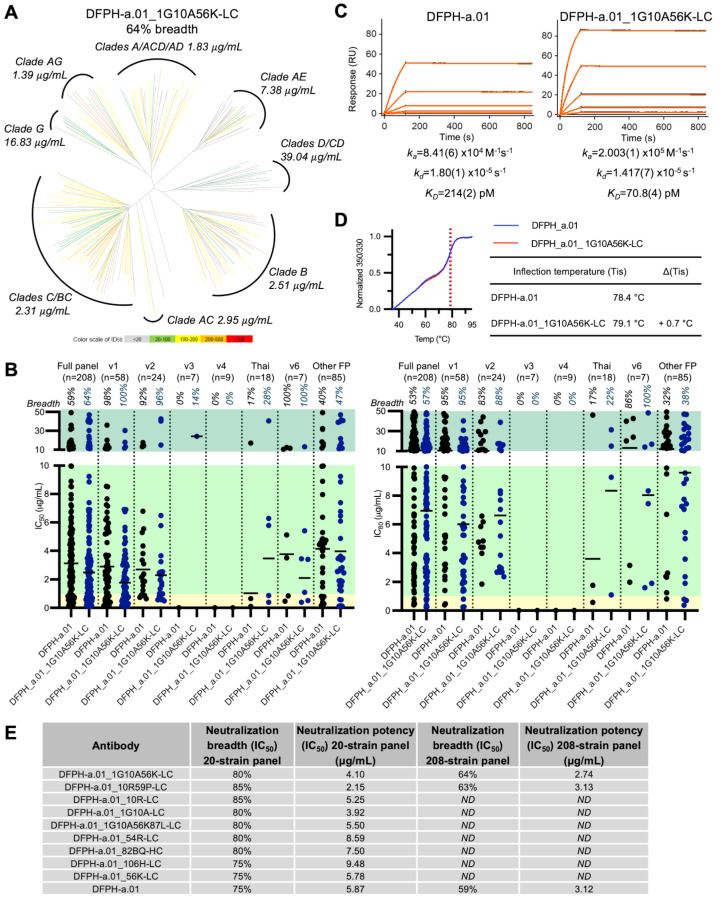
Antibody DFPH-a.01_1G10A56K-LC: neutralization breadth, Env trimer affinity, stability, and comparison of DFPH-a.01 antibodies. (**A**) Phylogenetic clustering revealed that DFPH-a.01_1G10A56K-LC recognized a wider range of HIV-1 clades and neutralized both DFPH-a.01-resistant and -sensitive variants more effectively. (**B**) Analysis of the 208-virus neutralization panel showed that the engineered antibody had expanded breadth relative to the original template across globally circulating HIV-1 strains. Viral strains are grouped by FP sequence. We used IC_50_ (**left**) or IC_80_ (**right**) of 50 µg/mL to determine neutralization sensitivity against a pseudovirus strain. (**C**) Surface plasmon resonance analysis of DFPH-a.01_1G10A56K-LC binding to BG505 Env trimers. Binding analysis was performed using BG505 DS-SOSIP tethered to the sensor chip surface via 2G12, with DFPH-a.01 and its mutant tested at five concentrations ranging from 1.11 to 90 nM. Black traces represent experimental data, and red lines represent the fit to the 1:1 interaction model. Numbers in brackets show the error of the fit to the last significant digit. (**D**) DFPH_a.01 and DFPH_a.01_1G10A56K-LC were assessed for stability by monitoring the change to the 350/330 ratio from intrinsic fluorescence of tryptophan during melting. (**E**) Summary of single- and multi-mutations with increased neutralization breadth compared to template DFPHa-01. ND, not determined. See also Appendix A.

## Data Availability

DNA sequences are deposited in NCBI GenBank as PX207628-PX207649. Resolved structure has been added to the PDB and EMDB databases as entries 9Q0W and EMD-72108, respectively. Code for analysis of immune receptor sequences is provided as part of Protocol 1: Clonal Variant Analysis of Antibody Engineering Libraries, https://github.com/dekoskylab/CSHL_protocols (accessed on 31 July 2025). For other items, any additional information required to reanalyze the data reported in this paper is available from the lead contact upon request. This study used the Office of Cyber Infrastructure and Computational Biology High Performance Computing cluster at the National Institute of Allergy and Infectious Diseases, Bethesda, MD, USA.

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
