# Peer review of "Yeast Display Reveals Plentiful Mutations That Improve Fusion Peptide Vaccine-Elicited Antibodies Beyond 59% HIV-1 Neutralization Breadth"

_vaccines, 2025, doi:10.3390/vaccines13111098_

Round 1
Reviewer 1 Report
Comments and Suggestions for Authors
This manuscript explores whether the neutralization breadth of fusion peptide (FP)-directed, vaccine-elicited antibodies against HIV-1 can be improved through systematic mutational analysis. Using yeast display and deep mutational scanning, the authors identified both single and multi-mutation variants that enhanced affinity, stability, and breadth compared with the parent antibody DFPH-a.01. Neutralization breadth increased from 59% to 63–64% across a 208-strain panel, with cryo-EM structures revealing that distal mutations influenced antibody conformation and antigen engagement. These findings demonstrate that beneficial mutations are plentiful and highlight potential strategies to optimize FP-directed vaccine responses. There are some points to be better clarified:
- While the improvements from 59% to 63–64% breadth are significant, please expand the discussion on whether such incremental gains are sufficient to change vaccine efficacy outcomes.
- Clarify how these insights could guide rational immunogen design or germline-targeting strategies.
- It would be valuable to speculate further on the potential upper limit of FP-directed antibody breadth. Is the ~64% observed here close to a ceiling, or might future engineering push this further toward the >80% range seen for some CD4bs-directed antibodies?
- The discussion focuses heavily on breadth; please expand discussion of potency improvements (IC50/IC80) and their relevance to protective titers.
- The site-saturation mutagenesis and multi-mutation library strategies are complex. Consider streamlining the methods description, possibly with a flow diagram summarizing library construction and screening workflow.
- Please acknowledge limitations, including the artificial nature of yeast display systems and potential differences compared to in vivo B cell evolution.
Author Response
Comment 1: This manuscript explores whether the neutralization breadth of fusion peptide (FP)-directed, vaccine-elicited antibodies against HIV-1 can be improved through systematic mutational analysis. Using yeast display and deep mutational scanning, the authors identified both single and multi-mutation variants that enhanced affinity, stability, and breadth compared with the parent antibody DFPH-a.01. Neutralization breadth increased from 59% to 63–64% across a 208-strain panel, with cryo-EM structures revealing that distal mutations influenced antibody conformation and antigen engagement. These findings demonstrate that beneficial mutations are plentiful and highlight potential strategies to optimize FP-directed vaccine responses. There are some points to be better clarified:
While the improvements from 59% to 63–64% breadth are significant, please expand the discussion on whether such incremental gains are sufficient to change vaccine efficacy outcomes.
Response 1: We thank the reviewer for this important point. We have expanded the Discussion (line 884) to clarify the biological relevance of incremental breadth gains. Specifically, we note that:
The high prevalence of mutations to improve DFPH-a.01 suggests the breadth/potency limit for DFPH-a class antibodies to be substantially higher than the 59% breadth of the current best vaccine-elicited antibody (DFPH-a.01). This is an important finding, as the breadth of the SHIV-infection boosted antibodies was at highest only 56% (for DJ85-a.01). Despite substantial in vivo affinity maturation (somatic hyper mutations - SHM - rates of 15-20%), the SHIV-infection boosted antibodies still did not appear to achieve that same level of maximal breadth. Likely these limits reflect the manner by which these antibodies were selected and matured and suggest that the immunization process can be further optimized to maximize FP-directed neutralization breadth. Even modest increases in breadth can also have meaningful biological implications when accompanied by corresponding potency gains. These findings suggest that raising the breath of DFPH-a-class antibodies, together with the potency improvements observed here, can combine to enhance protective efficacy.
Comment 2: Clarify how these insights could guide rational immunogen design or germline-targeting strategies.
Response 2: We added a new paragraph to the Discussion (Line 1318) highlighting how yeast-display-identified mutations reveal structural pathways by which FP antibodies improve:
Our mutational analyses also offer insights for rational immunogen design and germline-targeting strategies. Distal framework mutations such as S10R-LC, which alters Fab elbow angle, and S59P-LC, which increases CDR loop flexibility, provide further evidence that non-paratope residues can strongly influence FP antigen recognition. These structural pathways suggest that immunogens could be designed to favor B cell lineages with the capacity to access such mutations, thereby shaping antibody maturation toward mutationally favorable frameworks. Indeed, rational immunogen design has demonstrated proof-of-principle for guiding antibody evolution through tailored antigen engineering [38-41]. Applying these principles to germline-targeting strategies may facilitate the elicitation of FP-directed antibodies with improved breadth and potency, and would synergize well with recent structure-based FP immunogen designs. Interestingly, the most improving mutations for the macaque FP-directed antibody DFPH-a.01 emerged at para-tope-distal framework positions, whereas beneficial mutations in human FP-directed antibodies (e.g., VRC34.01) were often concentrated at or near the paratope. These differences may reflect distinct starting points for affinity maturation, and/or differences in the total duration of antigen exposure. In the macaque lineage, DFPH-a.01 began with relatively low baseline affinity and breadth, such that distal framework changes altering Fab elbow angle (S10R-LC) and CDR loop flexibility (S59P-LC) created structural preconditions that expanded recognition of diverse FP variants. In contrast, human FP bNAbs may have al-ready achieve high degrees of paratope complementarity, so further improvements were driven by fine-tuning residues proximal to the antigen contact surface. These differences highlight that multiple mutational trajectories, either distal framework remodeling or direct paratope optimization, can support the evolution of increased breadth and potency in FP-directed bNAbs.
Comment 3: It would be valuable to speculate further on the potential upper limit of FP-directed antibody breadth. Is the ~64% observed here close to a ceiling, or might future engineering push this further toward the >80% range seen for some CD4bs-directed antibodies?
Response 3: We appreciate this suggestion. In the revised Discussion (Line 897), we now explicitly state that the 63–64% breadth achieved here might not represent a ceiling for FP-directed antibodies:
Overall, yeast display revealed that many mutations can improve the neutralization breadth of DFPH-a.01, the best of the DFPH-a class of ‘reproducible’ or ‘multi-donor’ antibodies. This suggests that the breadth limit for this class of FP-directed antibodies is considerably higher than 60%, and the 63–64% breadth achieved here is unlikely to represent a ceiling for FP-directed antibodies. These yeast display efforts revealed panels of additional beneficial mutations, many of which were not fully evaluated in this study, suggesting that additional engineering would continue to enhance both potency and breadth. Given the high conservation of the FP across HIV-1 isolates and the repeated elicitation of FP-directed lineages in multiple animals, there appears to be substantial evolutionary space for additional improvements. Nevertheless, whatever the breadth limit, the breadth of FP-directed antibodies engineered here already exceeds the target 50% level, which has been suggested as a minimal breadth level that would lead to an impactful vaccine.
Comment 4: The discussion focuses heavily on breadth; please expand discussion of potency improvements (IC50/IC80) and their relevance to protective titers.
Response 4: We appreciate this insightful comment. In the revised Discussion (Line 874), we expanded our analysis to emphasize potency improvements alongside breadth:
Beyond breadth, our data also demonstrate meaningful improvements in potency. The DFPH-a.01_10R59P-LC variant showed ~2.7-fold enhanced IC50 potency in the 20-virus panel, and more modest but consistent potency gains across the 208-virus panel. These improvements are biologically meaningful, as macaque passive transfer studies indicate that serum ID50 (inhibitory dose) titers of ≥1:100 are sufficient to prevent SHIV acquisition [10]. Even modest increases in antibody potency can lower the threshold con-centration required for protection, enhancing the likelihood that vaccine-elicited antibodies achieve protective titers in vivo. Together, the combined improvements in breadth and potency highlight the translational impact of engineering FP-directed antibodies for vaccine development.
Comment 5: The site-saturation mutagenesis and multi-mutation library strategies are complex. Consider streamlining the methods description, possibly with a flow diagram summarizing library construction and screening workflow.
Response 5: We streamlined the Methods section (Sections 2.1–2.3) to reduce redundancy and improve clarity, and we added a new workflow schematic as suggested (Figure S1).
Comment 6: Please acknowledge limitations, including the artificial nature of yeast display systems and potential differences compared to in vivo B cell evolution.
Response 6: We added a paragraph in the Discussion section (Line 1342) addressing limitations of yeast display:
We note that yeast display is a system designed to explore protein-protein interaction affinities, and it does not fully recapitulate in vivo B cell evolution. SHM and selection occur inside germinal centers under complex intraclonal interaction dynamics based on factors such as antigen availability, competition among B cell clones, and the availability of T cell help. As a result, some mutations identified here may not be readily accessible to natural repertoires. Nevertheless, yeast display remains a powerful tool for uncovering structural mechanisms and mutational pathways to inform the next steps in rational vaccine design.
Reviewer 2 Report
Comments and Suggestions for Authors
Major comment
1.Biological relevance of breadth improvements: While the improvements from 59% to 63–64% breadth are statistically measurable, the manuscript should more clearly discuss whether such increments are biologically meaningful in the context of vaccine efficacy.
2.The study combines yeast display, large-scale mutational scanning, next-generation sequencing, virus neutralization panels, SPR, stability assays, and cryo-EM structural analysis. This breadth of methodology strengthens the robustness of the conclusions.
3.The identification of paratope-distal mutations (S10R, S59P) that modulate elbow angle and CDR positioning provides mechanistic understanding of how distal framework mutations can enhance neutralization.
4.Lack of reproducible screening/selection criteria. In Section 2.2 (Yeast Display LibraryScreening), the quantitative thresholds for classifying "low/medium/high-affinity fractions" (e.g., fluorescence signal cutoffs) are not specified.
5.Incomplete 208-virus panel data: Sections 3.4 and 3.7 only report aggregate neutralization breadth (63%/64%) of the mutant antibodies but lack breakdowns by HIV-1 clades or fusion peptide variants, hindering comprehensive evaluation of cross-clade efficacy.
Minor comment
1.The introduction is lengthy and could be streamlined by reducing redundancy about FP-directed antibodies from macaques.
2.Several acronyms (e.g., SSM, ER) are defined late in the methods—consider introducing them earlier.
3.“questions remain, however, as to whether their neutralization breadth and potency were sufficient to prevent HIV-1 infection.”Incorrect past tense “were” in the context of an ongoing scientific question.Revise to “...are sufficient to prevent HIV-1 infection.”
4.The “CO2” in section 2.4 should be written as “CO2”.
5.The “0.5x106 cells/mL” in section 2.5.1 should be written as “0.5x106 cells/mL”.
6.Some sentences are quite long and could be broken into shorter sentences for better readability.
Author Response
Major comment
Comment 1: Biological relevance of breadth improvements: While the improvements from 59% to 63–64% breadth are statistically measurable, the manuscript should more clearly discuss whether such increments are biologically meaningful in the context of vaccine efficacy.
Response 1: We appreciate this important point. In the revised Discussion, we now emphasize that even modest increases in breadth can be biologically meaningful when accompanied by potency gains. For example, DFPH-a.01_10R59P-LC achieved ~2.7-fold improved IC50 potency on the 20-virus panel, and modest potency improvements across the 208-virus panel. Prior macaque passive-transfer studies indicate that serum ID50 titers of ≥1:100 are protective; thus, increasing breadth from 59% to 63–64%, together with the potency gains, may provide a substantial boost to vaccine efficacy.
Comment 2: The study combines yeast display, large-scale mutational scanning, next-generation sequencing, virus neutralization panels, SPR, stability assays, and cryo-EM structural analysis. This breadth of methodology strengthens the robustness of the conclusions.
Response 2: We thank the reviewer for highlighting these aspects of the manuscript.
Comment 3: The identification of paratope-distal mutations (S10R, S59P) that modulate elbow angle and CDR positioning provides mechanistic understanding of how distal framework mutations can enhance neutralization.
Response 3: We have included this point in the Discussion.
Comment 4: Lack of reproducible screening/selection criteria. In Section 2.2 (Yeast Display LibraryScreening), the quantitative thresholds for classifying "low/medium/high-affinity fractions" (e.g., fluorescence signal cutoffs) are not specified.
Response 4: The thresholds used to define low-, medium-, and high-affinity gates in our yeast display screening are provided in the example flow cytometry plots and are drawn based on such quantitative gating examples and in combination with careful controls. We cannot provide a fluorescence signal cutoff, because it would change from day-to-day. The use of example gates along with careful controls is a well-accepted method for drawing reproducible and consistent flow cytometry sort gates, which is also robust to day-to-day instrument drift.
Comment 5: Incomplete 208-virus panel data: Sections 3.4 and 3.7 only report aggregate neutralization breadth (63%/64%) of the mutant antibodies but lack breakdowns by HIV-1 clades or fusion peptide variants, hindering comprehensive evaluation of cross-clade efficacy.
Response 5: We agree that cross-clade efficacy is important to delineate, and Figures 3A and 7A provide a view of cross-clade efficacy for DFPH-a.01 mutants. Detailed results of the 208-virus panel are provided in the Supplementary Materials (Tables S3–S4 and Figures S3, S5, S7). These include HIV-1 clade information and FP sequence variant, complementing the aggregate breadth values reported in the main text.
Minor comment
Comment 6: The introduction is lengthy and could be streamlined by reducing redundancy about FP-directed antibodies from macaques.
Response 6: We appreciate this helpful suggestion. We streamlined the discussion of FP-directed antibodies from macaques to reduce redundancy and improve readability. Specifically, we condensed overlapping descriptions of prior macaque vaccination and SHIV-boosting studies into a single concise paragraph, while retaining key details on breadth, potency, and protective titers. This revision highlights the relevance of macaque FP-directed antibodies, without repeating background information, and shortens the Introduction for improved clarity.
Comment 7: Several acronyms (e.g., SSM, ER) are defined late in the methods—consider introducing them earlier.
Response 7: Thank you. We reviewed and defined acronyms at their first appearance in the text.
Comment 8: “questions remain, however, as to whether their neutralization breadth and potency were sufficient to prevent HIV-1 infection. ”Incorrect past tense “were” in the context of an ongoing scientific question. Revise to “...are sufficient to prevent HIV-1 infection.”
Response 8: We have made this correction.
Comment 9: The “CO2” in section 2.4 should be written as “CO2”.
Response 9: Corrected throughout the text.
Comment 10: The “0.5x106 cells/mL” in section 2.5.1 should be written as “0.5x106 cells/mL”.
Response 10: Corrected.
Comment 11: Some sentences are quite long and could be broken into shorter sentences for better readability.
Response 11: We revised the entire manuscript text for readability.
Reviewer 3 Report
Comments and Suggestions for Authors
Broadly neutralizing antibodies (bNAbs) against HIV-1 target the prefusion-closed Env trimers and are categorized into several groups based on their target sites (V1/V2, V3 glycan, CD4-binding site, gp120/gp41 interface, fusion peptide, membrane proximal external region, and so on). They were isolated from HIV-1-positive elite controllers and observed, although not frequently, in vaccine-immunized human subjects and animals. The bNAbs are expected to utilize for HIV-1 treatment and prevention as an innovative strategy.
This study focuses on fusion peptide-directed macaque bNAb, DFPH-a.01, and increased its affinity and breadth through site saturation mutagenesis and yeast display library screening. The authors obtained a DFPH-a.01 variant, DFPH-a.01_10R59P-LC with S10R and S59P mutations in the VL chain and demonstrated higher neutralization potency and breadth (using 20-virus panel and 208-strain pseudovirus) with improved binding affinity to Env trimers (by SPR assay) and physical stability. To understand the molecular mechanisms of this improvement, they performed Cyro-EM and found that the mutations were far away from the paratope: the S10R mutation twisted the elbow of Fab and the S59P mutation increased the CDR loop flexibility. Lastly, they compared the paratope distance and mutation frequencies of the bNAbs for macaque, human, and mouse, all of which were improved by yeast display screening.
Overall, the data presented are clear and interesting. However, the supplementary figures/tables cited in the manuscript cannot be found in the manuscript. No additional files were seen.
The mutations improving macaque bNAb emerged at the distal positions from the paratope in the light chain (in this study). In contrast, the mutations improving human bNAb are accumulated at or near the paratope. I would like the authors to add a discussion about why these differences appeared. Such accumulation of mutations at or near the paratope is possibly reflected in significant increase in breadth (from 60% to 80%) and potency (by 10-fold).
Minor:
1) Figure panels and legends: Panels in figures are represented by uppercase but are described in lowercase in legends.
2) Figure 8B: the S43K mutation on the vFP16.02 VL line is perhaps S48K.
Author Response
Broadly neutralizing antibodies (bNAbs) against HIV-1 target the prefusion-closed Env trimers and are categorized into several groups based on their target sites (V1/V2, V3 glycan, CD4-binding site, gp120/gp41 interface, fusion peptide, membrane proximal external region, and so on). They were isolated from HIV-1-positive elite controllers and observed, although not frequently, in vaccine-immunized human subjects and animals. The bNAbs are expected to utilize for HIV-1 treatment and prevention as an innovative strategy.
This study focuses on fusion peptide-directed macaque bNAb, DFPH-a.01, and increased its affinity and breadth through site saturation mutagenesis and yeast display library screening. The authors obtained a DFPH-a.01 variant, DFPH-a.01_10R59P-LC with S10R and S59P mutations in the VL chain and demonstrated higher neutralization potency and breadth (using 20-virus panel and 208-strain pseudovirus) with improved binding affinity to Env trimers (by SPR assay) and physical stability. To understand the molecular mechanisms of this improvement, they performed Cyro-EM and found that the mutations were far away from the paratope: the S10R mutation twisted the elbow of Fab and the S59P mutation increased the CDR loop flexibility. Lastly, they compared the paratope distance and mutation frequencies of the bNAbs for macaque, human, and mouse, all of which were improved by yeast display screening.
Comment 1: Overall, the data presented are clear and interesting. However, the supplementary figures/tables cited in the manuscript cannot be found in the manuscript. No additional files were seen.
The mutations improving macaque bNAb emerged at the distal positions from the paratope in the light chain (in this study). In contrast, the mutations improving human bNAb are accumulated at or near the paratope. I would like the authors to add a discussion about why these differences appeared. Such accumulation of mutations at or near the paratope is possibly reflected in significant increase in breadth (from 60% to 80%) and potency (by 10-fold).
Response 1: We thank the reviewer for this careful summary of our study and the constructive suggestions.
Missing supplementary figures/tables: The supplementary figures and tables have been uploaded and cross-referenced correctly in the revised manuscript (see Supplementary Figures S1–S9 and Supplementary Tables S1–S5). These materials provide the full data underlying the text.
Differences between macaque and human bNAb mutation patterns: We agree this is an important point. In the revised Discussion (Line 1318), we added a new paragraph addressing this point.
Minor
Comment 2: Figure panels and legends: Panels in figures are represented by uppercase but are described in lowercase in legends.
Response 2: Thanks. We have corrected this.
Comment 3: Figure 8B: the S43K mutation on the vFP16.02 VL line is perhaps S48K.
Response 3: Thank you, this difference is a result of the Kabat vs. template numbering systems used (S48K in template numbering is equivalent to S43K in Kabat numbering). We clarified the text on this point.
Reviewer 4 Report
Comments and Suggestions for Authors
Very thorough, clearly presented and informative. Nice paper
Author Response
Comment 1: Very thorough, clearly presented and informative. Nice paper
Response 1: Thank you!
Round 2
Reviewer 2 Report
Comments and Suggestions for Authors
No more comments for the authors. The authros have answered all the concerns.